# Inflammatory Cells Can Alter the Levels of H3K9ac and γH2AX in Dysplastic Cells and Favor Tumor Phenotype

**DOI:** 10.3390/jpm13040662

**Published:** 2023-04-13

**Authors:** Camila de Oliveira Barbeiro, Darcy Fernandes, Mariana Paravani Palaçon, Rogerio Moraes Castilho, Luciana Yamamoto de Almeida, Andreia Bufalino

**Affiliations:** 1Oral Medicine, Department of Diagnosis and Surgery, School of Dentistry, São Paulo State University (Unesp), Araraquara 14801-903, SP, Brazil; camila.barbeiro@unesp.br (C.d.O.B.);; 2Laboratory of Epithelial Biology, Department of Periodontics and Oral Medicine, University of Michigan, 1011N University Av, Ann Arbor, MI 48109-1078, USA

**Keywords:** co-culture, epithelial dysplasia, malignant transformation, peripheral blood mononuclear cell

## Abstract

Oral potentially malignant disorders (OPMD) are clinical presentations that carry an increased risk of cancer development. Currently, epithelial dysplasia grade is based on architectural and cytological epithelial changes and is used to predict the malignant transformation of these lesions. However, predicting which OPMD will progress to a malignant tumor is very challenging. Inflammatory infiltrates can favor cancer development, and recent studies suggest that this association with OPMD lesions may be related to the etiology and/or aggressive clinical behavior of these lesions. Epigenetic changes such as histone modifications may mediate chronic inflammation and also favor tumor cells in immune resistance and evasion. This study aimed to evaluate the relationship between histone acetylation (H3K9ac) and DNA damage in the context of dysplastic lesions with prominent chronic inflammation. Immunofluorescence of “low-risk” and “high-risk” OPMD lesions (*n* = 24) and inflammatory fibrous hyperplasia (*n* = 10) as the control group was performed to assess histone acetylation levels and DNA damage through the phosphorylation of H2AX (γH2AX). Cell co-culture assays with PBMCs and oral keratinocyte cell lines (NOK-SI, DOK, and SCC-25) were performed to assess proliferation, adhesion, migration, and epithelial–mesenchymal transition (EMT). Oral dysplastic lesions showed a hypoacetylation of H3K9 and low levels of γH2AX compared to control. The contact of dysplastic oral keratinocytes with PBMCs favored EMT and the loss of cell–cell adhesion. On the other hand, p27 levels increased and cyclin E decreased in DOK, indicating cell cycle arrest. We conclude that the presence of chronic inflammation associated to dysplastic lesions is capable of promoting epigenetic alterations, which in turn can favor the process of malignant transformation.

## 1. Introduction

The presence of oral potentially malignant disorders (OPMD) indicates an increased risk of developing cancer of the lip or oral cavity during the patient’s lifetime; nonetheless, only a small proportion progresses to cancer [1]. Considering all OPMD, the overall rate of malignant transformation was 7.9% [2]. Histopathological analysis of OPMD biopsies may show evidence of epithelial dysplasia that provides important information regarding the prognosis of the condition [3,4]. However, predicting which OPMD lesions will in fact progress to a malignant tumor is very challenging.

Epithelial dysplasia is a spectrum of architectural and cytological epithelial changes caused by the accumulation of genetic alterations and is associated with an increased risk of progression to oral squamous cell carcinoma (OSCC) [1,4]. The establishment of epithelial dysplasia grade in OPMD is based on the individual epithelial cellular features and the architectural changes [4]. Consequently, the term epithelial dysplasia is applied to a lesion in which part or all of the epithelial thickness is replaced by cells showing varying degrees of cellular atypia [5]. In this sense, the three-grade system and binary system are currently the main proposed applied tools to predict malignant transformation in OPMD [4].

Epigenetic events are responsible for regulating gene expression without altering the nucleotide sequence of DNA, and it can be observed during malignant transformation and carcinogenesis [5]. Histone acetylation is one of the most studied epigenetic events and appears to influence several important cellular processes, such as cell cycle progression, DNA repair, and apoptosis [6,7,8]. Immune responses can suppress tumorigenesis but also contribute to cancer initiation and progression, suggesting a complex interaction between the immune system and cancer. Cancer cells utilize the epigenetic silencing of immune-related genes to evade the immune response. Chronic inflammatory lesions such as oral lichen planus, leukoplakia, Barrett’s esophagus, and inflammatory bowel diseases are associated with malignant transformation [9]. Thus, it seems important to study the participation of the inflammatory infiltrate in cellular processes of dysplastic lesions with the potential to undergo malignant transformation.

In this context, the immune system can identify and destroy pre-neoplastic cells in a process called cancer immunosurveillance, which functions as an important defense against cancer development [10,11,12]. Recently, data obtained from investigations in cancer mouse models and humans with cancer offer compelling evidence that certain immune cell types, effector molecules, and specific pathways of innate and adaptive immunity can sometimes function collectively as extrinsic tumor-suppressive mechanisms. However, the immune system can also promote tumor progression through a mechanism called escape, in which pre-cancerous or cancerous cells foster immune evasion [10,13,14]. In view of these findings, the concept of cancer immuno-editing emerged. It comprises the elimination phase, in which the immune system recognizes and kills the potentially malignant cells; the equilibrium phase, in which an immune selection of tumor cells with reduced immunogenicity occurs; and the escape phase of tumor development, in which the immune system acts favoring tumor progression [10].

The concept of tumor immunosurveillance assumes that the immune system can recognize potentially malignant cells and, in most cases, destroy them before they become clinically apparent. If immunosurveillance plays an important role in tumor suppression, then we might assume that patients with OPMDs would exhibit vigorous immune responses to eliminate these abnormal cells. On the other hand, the inflammatory infiltrate may also impact oral keratinocytes, resulting in alterations that may favor tumor phenotypes and the malignant transformation of OPMD. Yet, studies evaluating the participation of the inflammatory infiltrate in the process of malignant transformation of dysplastic lesions are scarce.

Thus, this study aimed to evaluate the relationship between histone acetylation (H3K9ac) and DNA damage in the context of dysplasia lesions with prominent chronic inflammation. In addition, it was evaluated whether the contact of dysplastic and tumor cell lines with peripheral blood mononuclear cells (PBMCs) impacts the levels of histone acetylation and DNA damage, cell proliferation, cell cycle, epithelial–mesenchymal transition (EMT), and cell migration, conferring advantages or control of tumorigenesis.

## 2. Materials and Methods

### 2.1. Ethics Statement

This study was conducted in accordance with the Declaration of Helsinki and approved by the local Ethics Committee (CAAE: 96976718.3.0000.5416). All human samples were retrieved from the archives of the Department of Diagnosis and Surgery, São Paulo State University (Unesp), School of Dentistry (Brazil), and all participants agreed to participate in the study and provided written informed consent.

### 2.2. Tissue Samples and Immunofluorescence Analysis

For this study, the sample size was calculated for all groups using the following parameters: minimum difference of means, standard deviation of the standard error, α = 0.05, test power, and number of treatments, which resulted in a minimum number of 4 samples for each group. Thus, histopathological slides of 24 biopsies from patients with dysplastic lesions were reassessed, and the presence and intensity of the inflammatory infiltrate were evaluated by two pathologists independently. For this analysis, 5 representative fields were evaluated in high resolution (400×), and the average value of each sample was determined. Subsequently, the samples were classified according to the potential susceptibility for malignant transformation into “low risk” and “high risk”, according to the binary system proposed by Kujan et al., 2006 [15]. In addition, a total of 10 samples of inflammatory fibrous hyperplasia (IFH) were used as a control group in the immunofluorescence assay, as these lesions exhibit many areas of chronic inflammation in the lamina propria and do not present potential for malignant transformation.

The immunofluorescence was performed in the sections which were initially dewaxed and hydrated. Antigen retrieval was performed using citric acid anhydrous (pH 6). After incubation with 1% bovine serum albumin (BSA) and 0.1% Triton in phosphate–saline buffer solution (PBS), the primary antibodies anti-histone H3K9—Acetyl-Histone H3 (Lys9) (C5B11, Rabbit mAb, Cell Signaling Technology, Danvers, MA, USA, Overnight, 1:400 dilution) and anti-phospho-Histone H2AX (Ser139) (JBW301, Mouse, EMD Millipore Corporation, Burlington, MA, USA, Overnight, 1:200 dilution) were added to the tissue samples, which was followed by FITC or TRITC-conjugated secondary antibody. Tissues were counterstained with Hoechst 33342 (Thermo Fisher Scientific, Waltham, MA, USA) and mounted using an aqueous mounting medium (Fluoroshield, Sigma-Aldrich, St. Louis, MO, USA, #F6182). Tissue reactivity in all cases was evaluated by analyzing the digital images captured with Nikon Eclipse 80i Microscope, (Nikon, Melville, NY, USA) and the software NIS-Element. Five consecutive fields of each case were captured at 200× magnification, and two independent and blind examiners used the Image J software to count positive cells in green. We performed morphometric image analysis for H3K9ac and γH2AX using the software ImageJ (Version 1.38s; NIH, Bethesda, MD, USA). All positive and negative cells were counted in each field, and the percentage of the total number of cells in each case was calculated. For H3K9ac, we also performed staining intensity analysis, which was quantified by Image J software. DAPI was used to count the total number of epithelial cells per field. The positive index was calculated as mentioned above.

### 2.3. Cell Culture

The NOK-SI cell line is spontaneously immortalized normal oral keratinocytes derived from human oral mucosa (NIDCR/NIH) and was provided by Dr. Carlos Rossa Junior and maintained in DMEM (Gibco, Cat# 11965)/10% fetal bovine serum (FBS) supplemented with penicillin and streptomycin [16]. In addition, commercial DOK and SCC-25 keratinocyte cell lines were used in all experiments. The DOK cell line (CVCL_1180; ECACC—European Collection of Authenticated Cell Cultures, London, UK), a human dysplastic oral keratinocyte cell line obtained from a white lesion of the tongue of a 57-year-old man with mild/moderate dysplasia, was cultivated in DMEM (Gibco) supplemented with L-Glutamine, 10% FBS, 5 μg/mL hydrocortisone, and 200 U/mL Penicillin/Streptomycin (Gibco). The human OSCC cell line, SCC-25, was obtained from American Type Culture Collection (ATCC, USA) and cultured as recommended in a 1:1 mixture of DMEM and Ham’s F12 medium (DMEM/F12; Invitrogen, Waltham, MA, USA) containing 1.2 g/L sodium bicarbonate, 2.5 mM L-glutamine, 15 mM HEPES and 0.5 mM sodium pyruvate supplemented with 10% FBS (Cultilab Ltd., Campinas, Brazil), 0.4 μg/mL hydrocortisone (Sigma-Aldrich, St. Louis, MO, USA) and antibiotic (Invitrogen, USA). All cell lines were maintained at 37 °C in an atmosphere containing 5% CO_2_.

### 2.4. Peripheral Blood Mononuclear Cells (PBMCs) Isolation

Peripheral blood was collected from healthy donors (without diabetes, syphilis, hepatitis, and HIV) using a vacuum system. The technique of Ficoll–Hypaque (GE Healthcare) density centrifugation of heparinized blood was used to obtain peripheral blood mononuclear cells (PBMCs). The PBMC composite sediment was resuspended in RPMI 1640 culture medium (Invitrogen, USA) supplemented with L-Glutamine, 10% fetal bovine serum (FBS; Cultilab Ltd., Brazil), 10 mM HEPES (4-(2-hydroxyethyl)-1-piperazineethanesulfonic acid) and 200 U/mL of Penicillin/Streptomycin (Invitrogen, USA).

### 2.5. In Vitro Study Design of Co-Culture

Co-culture assays were performed to assess the effects of direct contact with PBMCs on the different keratinocyte lines. After PBMCs isolation, they were harvested in co-culture with the different cell lines in a proportion of 1 (PBMCs): 5 (NOK-SI, DOK, or SCC-25) for different periods, according to the experimental outcome of interest. Each cell line was divided into two groups, with contact and without contact with PBMCs. In the co-culture groups, leukocytes from peripheral blood were removed through Dynabeads™ CD45 (ThermoFisher Scientific, USA), as indicated by the manufacturer. 

### 2.6. Cell Proliferation and Cell Cycle Assay by Flow Cytometry

The different oral keratinocytes were cultivated in co-culture with or without PBMCs, as described above. The different cell lines of keratinocytes were cultivated on 6-well plates at 1.5 × 10^5^ cells/well with culture medium supplemented with 10% FBS and maintained at 37 °C in an atmosphere containing 5% CO_2_. After 24 h, the culture medium was changed to fresh and not supplemented with FBS to promote cell synchronization and then cultivated for 48 h. After this period, PBMCs were added to the designed groups to evaluate the effect of co-culture with the keratinocytes, and all groups received a fresh culture medium with 1% FBS. After 24 h, the cells were washed with PBS, trypsinized, and centrifuged at 350× *g* for 5 min. The sediment was resuspended in PBS, which was followed by further centrifugation. For proliferation assay, the cells were fixed and permeated with 70% ethanol, drop by drop, under agitation for 30 s, which was followed by incubation for 1 h in the dark at −20 °C. Subsequently, the cells were centrifuged and washed twice with 1% FBS buffer diluted in PBS. At the end of the washes, approximately 1.5 × 10^5^ cells were marked with 5 µL of the primary antibody PE-CyTM7 Mouse anti-Ki-67 (Clone B56, BD Pharmingen) and incubated in a dark room for 30 min at room temperature. After incubation, the precipitate was washed twice and resuspended in 1% diluted PBS buffer. For the cell cycle analysis, the cells were fixed in 70% cold ethanol for 12 h at −20 °C and then stained with 50 µL/mL of PI (Sigma-Aldrich, USA) for 1 h in the dark at 4 °C. A minimum of 10,000 events were acquired in the Fortessa LSR flow cytometer (BD Biosciences, Franklin Lakes, NJ, USA) and the proliferation data were analyzed in the FlowJo software (version 10.0.7, Tree Star), and the results were expressed by the median fluorescence intensity (MFI). To determine the percentage of cells in each cell cycle phase, we used The ModFit LTTM program (BD Biosciences, USA).

### 2.7. RT-qPCR Analysis of Epithelial-Mesenchymal Transition (EMT)

The gene expression of EMT markers was evaluated in the different oral keratinocytes cell lines cultivated in 25 cm^2^ flasks with or without PBMCs, at 37 °C in an atmosphere containing 5% CO_2_. After 24 h, the keratinocytes were washed with PBS, trypsinized, and centrifuged at 400 g for 3 min. The RNA was extracted using TRIzol reagent (Invitrogen, USA). The total RNA concentration was determined by absorbances with 260 nm and 280 nm wavelengths (Nanodrop 2000c spectrophotometer, Thermo Fisher Scientific, USA). Reverse transcription was performed using the High-Capacity cDNA Archive kit (Applied Biosystems, USA), according to the manufacturer’s instructions. RT-qPCR reaction was performed using 5 ng of diluted cDNA, 5 pmole of each primer (Appendix A), and GoTaq qPCR^®^ Master Mix in a final volume of 10 μL (Promega, Madison, WI, USA).

### 2.8. Western Blot

Western blot analysis was used to determine the expression of proteins related to the cell cycle to confirm the effects of PBMCs on EMT markers and also to investigate the levels of H3K9 acetylation and phosphorylation of H2AX. Co-cultures of PBMCs and the different cell lines of keratinocytes were grown in 100 mm^2^ plates with 70% to 80% confluence and then washed with cold PBS, and subsequently, the cells were detached from the plates with a cell scraper (Costar, Washington, DC, USA). After centrifugation, the cell precipitates were incubated with 100 μL of lysis buffer (Roche, Grenzach-Wyhlen, Germany), and protein concentrations were measured using the Bradford method (Bio-Rad, Hercules, CA, USA). Standardized amounts of total protein for each antibody reaction were electrophoretically separated in a polyacrylamide gel with 10% sodium dodecyl sulfate (SDS-PAGE) under reducing conditions and transferred to nitrocellulose membranes (Millipore, MA, USA). The membranes were blocked with 5% fat-free milk and then incubated with primary antibodies (Appendix A) followed by a secondary peroxidase-conjugated antibody. After washing, the protein bands were detected using an enhanced chemiluminescence (ECL) Western Blotting System (Bio-Rad Laboratories, Hercules, CA, USA).

### 2.9. Cell Migration

To analyze the influence of PBMCs on cell migration, a scratch assay was performed [17]. The different cell lines were grown under normal nutrition conditions in 6-well plates until 95% confluence. Then, an in vitro wound (scratch) was performed with a 200 μL pipette tip, the medium was aspirated, and cells were washed and incubated with complete media supplemented with FBS. The keratinocytes (NOK-SI, DOK, or SCC-25) were incubated in the presence or absence of PBMCs, and digital images of three microscopic fields (40× magnification) were obtained from each well at 0, 3, 6, 9, 12, 15, and 18 h using a Leica DM 2500 trinocular inverted microscope (Leica Microsystems). The wound area was analyzed using the Image J software, and the percentage of healing at different times was calculated in relation to the total wound area at time 0 h from the same wound site.

### 2.10. Hanging Drop Assay

To evaluate the participation of PBMCs in the regulation of cell–cell contact of the different cell lines (NOK-SI, DOK, or SCC-25), the hanging drop assay was performed as previously described [18]. Briefly, 2 × 10^4^ keratinocytes alone or in co-culture with PBMCs were resuspended in 27 µL drops of complete media and kept in the lid of a 100 mm^2^ plate for 16 h at 37 °C in a 5% CO_2_ air atmosphere. Images of cell aggregates of 5 random fields from 5 different suspensions were visualized on a Leica DM 2500 trinocular inverted microscope (Leica Microsystems, Wetzlar, Germany).

### 2.11. Statistical Analysis

All in vitro experiments were repeated at least three times and expressed as mean ± standard deviation. The data distribution was determined by the Shapiro–Wilk normality test for the subsequent application of appropriate statistical tests according to the data distribution. For those assays, the Mann–Whitney U test or one-way analysis of variance (ANOVA) with post hoc comparisons based on Games–Howell’s multiple comparisons test was applied. When appropriate, some results were also evaluated descriptively. The level of significance considered was 5% (*p* = 0.05), and statistical analysis was performed using Prism 4, GraphPad Software Inc. (San Diego, CA, USA).

## 3. Results

### 3.1. Oral Dysplastic Lesions Show Hypoacetylation of H3K9 and Low Levels of γH2AX

In this study, we showed that the frequency of subepithelial inflammatory cells was significantly higher in samples classified as “high risk” compared to “low risk” samples (95% CI: 0.000 to 93.20; *p* = 0.006), predominantly presenting a chronic lymphoplasmacytic inflammatory infiltrate (Figure 1). Furthermore, it has been demonstrated through immunofluorescence that dysplastic oral lesions exhibit hypoacetylation of H3K9 and low levels of γH2AX compared to the control group (IFH). Considering the H3K9ac intensity of labeling, there was a statistically significant reduction in H3K9ac levels between the “low-risk” (95% CI: 2854.89 to 6274.81; *p* < 0.001) and “high-risk” groups (95% CI: 467.65 to 3993.28; *p* = 0.01) compared to the control group as well as a significant reduction in H3K9ac levels between the low-risk and the high-risk group (95% CI: −3549.90 to −1118.87; *p* < 0.001). On the other hand, the percentage of H3K9ac positive cells in the intraepithelial area was lower in the “low-risk” group compared to the control group (95% CI: −30.61 to −4.45; *p* = 0.006). In addition, H2AX phosphorylation levels decreased in “high-risk” lesions when compared to “low-risk” lesions (95% CI: −0.76 to −0.06; *p* = 0.015) and control (95% CI: −1.50 to −1.97; *p* = 0.009). Figure 2 shows the representative images of H3K9ac and γH2AX immunofluorescence and graphics of quantitative analysis.

### 3.2. PBMCs Do Not Influence the Proliferation of Oral Dysplastic and Neoplastic Cells

The exposure of oral keratinocyte lineages (NOK-SI, DOK, or SCC-25) to PBMCs for 48 h could not significantly influence the proliferation potential of these cells as observed in the relative MIF of Ki-67 expression (Figure 3A). Likewise, there were no statistically significant differences in the distribution of the cells in the different phases of the cell cycle (Figure 3B). However, the accumulation of cells in phases G0/G1 showed an upward trend. Therefore, the production of several proteins involved in the G1 phase of the cell cycle was evaluated through Western blot. An expressive increase in p27^Kip1^ levels was observed in the cell lineages NOK-SI, DOK, and SCC-25 after co-culture with the PBMCs, with this increased expression being more evident in the DOK cell line. In addition, there were no significant changes in p16^INK4^ and p21^WAF1/Cip1^ levels in the NOK-SI, DOK, or SCC-25 lineages. The levels of cyclin-dependent kinases (Cdks) and cyclins were also evaluated to verify the proteins and enzymes involved in cell cycle progression. The synthesis of cyclin E and cyclin D decreased in the co-culture of PBMCs with DOK and SCC-25, respectively. Simultaneously to protein levels analysis involved in the cell cycle, the levels of H3K9 acetylation (H3K9ac) and H2AX phosphorylation were also evaluated utilizing Western blot. The results showed an increase in H3K9ac levels in the SCC-25 lineage in co-culture with PBMCs and a slight reduction in the DOK group in co-culture with PBMCs. In addition, a reduction in H2AX phosphorylation levels in all keratinocyte lineages cultivated with PBMCs was observed. These results indicate a reduction in the levels of DNA damage in all the oral keratinocyte lineages studied in the presence of PBMCs. Figure 3C shows representative images of a Western blot assay to check the protein content of proteins involved in cell cycle control, H3K9ac, and H2AX phosphorylation, using total protein extract from keratinocyte lineages with and without contact with PBMCs.

### 3.3. Direct Co-Culture between PBMCs and Dysplastic Cells May Play a Role in EMT and Loss of Cell-Cell Adhesion

To evaluate whether PBMCs influence differently the EMT of the dysplastic keratinocyte lineages compared to the tumor and control group, the cadherin switching was examined by RT-qPCR and Western blot. Although no statistically significant difference in the levels of cadherins mRNA was identified, it was possible to observe a tendency in the suppression of E-cadherin and an increase in the expression of N-cadherin in the co-culture of DOK with PBMCs. In addition, the cadherin switch was observed by Western blot analysis in the same group after 48 h of contact with PBMCs. The SCC-25 lineage presented high levels of the two markers in both experimental conditions, whereas N-cadherin levels increased slightly after co-culture with the PBMCs. Furthermore, the SCC-25 lineage revealed high levels of N-cadherin expression compared to NOK-SI and DOK cells irrespective of contact with PBMCs.

Due to the slight changes in the expression of EMT markers, the hanging drop assay was performed to evaluate possible modulations in the cell–cell adhesion of oral keratinocyte lineages after contact with PBMCs. In this assay, it was possible to observe the formation of large cell aggregates in the NOK-SI group, while the DOK and SCC-25 cells revealed the formation of small cell aggregates before contact with PBMCs. However, the DOK and SCC-25 cell lines showed a change in the pattern of formation of these aggregates after exposure to PBMCs. We note that the presence of PBMCs was able to promote the complete deconstruction of the DOK and SCC-25 cell aggregate compared to the control group not exposed to PBMCs. Figure 4 shows the impact of direct co-culture between PBMCs and different keratinocyte lineages on the cadherin switching and capacity of cells to engage in cell–cell.

### 3.4. Direct Co-Culture between PBMCs and Dysplastic Cells Does Not Interfere with the Process of Cell Migration

Cell migration is an essential phenomenon for the invasion and metastasis process. When a wound is made in a monolayer of cells, the cells with high migratory capacity cross this line, making it possible to study drugs that may interfere in this mechanism. To visualize changes in the migratory capacity of NOK-SI, DOK, and SCC-25 cell lines in the absence or presence of PBMCs, we performed a comparative analysis using the scratch assay method (Figure 5A). As a result, the presence of PBMCs was able to reduce the wound of NOK-SI and SCC-25 cell lines. Statistical analysis showed a statistically significant increase in repair rates in these two groups following the 15 h of cell co-culture (Figure 5B).

## 4. Discussion

To evaluate epigenetic changes in oral dysplastic lesions involving histones and indirectly analyze the accumulation of DNA damage, immunofluorescence analysis through H3K9ac and γH2AX was performed. Having observed a possible association between the dysplasia degree and histone modification and knowing that there is an inflammatory infiltrate in these lesions, we hypothesized that perhaps the dysplastic cells modulate the inflammatory microenvironment to favor malignant transformation. Therefore, we performed an in vitro study to assess if the inflammatory cells in co-culture with oral keratinocytes are capable of promoting the same epigenetic changes found in the immunofluorescence analysis. In addition, we evaluated if the interaction between PBMCs and dysplastic cells influences the main phenotypes associated with tumorigenesis, such as cell proliferation, EMT, cell–cell adhesion, and migration.

Epigenetic alterations such as the imbalance of histone acetylation in promoter regions contribute to the dysregulation of gene expression and have been associated with carcinogenesis and cancer progression. The hyperphosphorylation of H2AX, hypo-, and hyperacetylation of H3K9 is more frequently observed in OSCC and OPMD [19]. Our results revealed differences in H3K9ac levels in the “low risk” and “high risk” groups compared to the control, showing that both “low risk” and “high risk” dysplastic lesions exhibit hypoacetylation of H3K9 compared to the control group. Elevated levels of H3K9ac were observed in oral lichen planus lesions that did not respond well to treatment and relapsed [20], and reduced levels of H3K9ac and H4K12ac as well as reduced methylation levels were observed in actinic cheilitis, suggesting a relationship with the onset of actinic damage but not essentially to malignant transformation [21]. Interestingly, these latter lesions compose the OPMD group, suggesting then that elevated H3K9ac levels may predict recurrent and unresponsive lesions, and conversely, reduced levels of H3K9ac may initiate together with the onset of epithelial dysplasia changes. Webber et al. (2017) also found high levels of H3K9ac in oral leukoplakia lesions with no statistical difference compared to normal mucosa. In contrast, they found hypoacetylation of H3K9ac in OSCC lesions, and it correlated with a worse prognosis [19]. This may suggest that the loss of histone H3K9 acetylation may occur as a late event in carcinogenesis.

γH2AX is a key player in the DNA damage response and repair machinery. When a double-stranded break (DSB) occurs, the histone 2A (H2A) protein is phosphorylated, forming the modified form γH2AX. This modification serves as a marker for DSBs, as it can be detected as punctate staining in the nucleus. The presence of γH2AX is a reliable indicator of DSBs, which can help guide the repair process. Furthermore, it is also involved in the recruitment of proteins involved in the repair process, such as DNA repair enzymes, DNA damage checkpoint proteins, and chromatin remodeling proteins [22]. A previous study revealed that in epithelial cells of the oral mucosa, it was not possible to detect phosphorylated γH2AX [23]. In addition, compared to OSCC, γH2AX expression levels were significantly increased in epithelial hyperplasia and dysplasia, being significantly elevated in the dysplasia relative to the other groups [23]. In contrast, our results showed that γH2AX levels in “low-risk” and “high-risk” lesions decreased compared to control, with “high-risk” lesions having the lowest level of γH2AX. However, different from our study, the authors did not use the binary grading system (high/low risk) of oral epithelial dysplasia, and the pattern of γH2AX labeling in the samples was not evaluated according to the degree of dysplasia. In addition, we evaluated the expression of γH2AX within a context of inflammation, using samples of inflammatory fibrous hyperplasia as a control group, which have an intense chronic inflammatory infiltrate, unlike normal oral mucosa. Thus, we can suggest that in the control and the “low-risk” groups, repair control is being activated more effectively, preventing abnormal cells from continuing to proliferate. In the “high-risk” group, however, this repair process is not being well recruited, which may justify the high rates of malignant transformation in this group, as described in the literature.

Cell cycle control is essential to ensure proper cell division and adequate proliferation. If there is deregulation in this control, there will be uncontrolled proliferation, enabling neoplastic growth. The progression of cell cycle phases occurs by the activation and inhibition of cyclin-dependent kinases (Cdks), which are controlled by the levels and availability of cyclins and CDK inhibitor proteins (cki) [24,25]. P16^Ink4a^, p21^WAF1/Cip1^, and p27^Kip1^ inhibit cell cycle progression by forming complexes with Cdks [26]. Our study revealed that the DOK cell line showed the lowest level of p27^Kip1^ expression in relation to the other cell line in the absence of PBMCs. The activation of p27 promotes cell cycle arrest and prevents altered cells from continuing to the next phases of the cell cycle. Reports in the literature show that low levels of p27 are present in prostate intraepithelial neoplasia, breast cancer, premalignant and malignant non-invasive lesions such as breast carcinoma in situ as well as cervical carcinoma and metastatic tumors. These findings indicate that decreased levels of p27 contribute to cancer progression and transition from carcinoma in situ to invasive carcinoma [27,28,29]. These findings are supported by other studies that demonstrate a correlation between p16, p27, and the malignant transformation of OPMDs. For example, one study showed that high risk OPMDs and OSCC lesions showed significantly lower expression of p16 and p27 compared to normal oral mucosa [30]. Furthermore, another study showed that p27 expression was significantly lower in OSCC tissues compared to normal and dysplastic oral mucosa [29]. Overall, these studies demonstrate that P27 and P16 contribute to carcinogenesis and tumor progression. However, a significant increase in p27^Kip1^ levels in NOK-SI, DOK, and SCC-25 cell lines after co-culture with PBMCs was observed. This result suggests that PBMCs stimulated the expression of a tumor suppressor gene probably because abnormal cells were detected in an in vitro condition. These findings lead us to consider that the influence of PBMCs on dysplastic and tumor keratinocyte lineages was positive in stimulating the expression of p27 and thus contributing to cell cycle control and preventing malignant transformation as well as enhancing tumor suppression in the SCC-25 lineage, indicating an anti-tumor influence of PBMCs on different keratinocytes and supporting the concept of immunosurveillance.

Additionally, we noticed that cyclin D and cyclin E levels decreased in DOK and SCC25 after the co-culture with PBMCs. These results indicate that the presence of the PBMCs modulated the production of some proteins involved in the G1/S transition of the cell cycle to halt the progression of the cell cycle, highlighting an anti-tumorigenic role of the PBMCs. The study by Guan et al. (2018) also evaluated cyclin D1 levels in dysplastic epithelial lesions and OSCC compared to normal epithelium. Epithelial dysplasia and OSCC showed increased levels of cyclin D1, with the highest levels seen in the OSCC group. Lesions with mild dysplasia showed lower levels of cyclin D1 than moderate dysplasia [29]. In our study, the co-culture of PBMCs with the SCC-25 cell line led to decreased cyclin D levels, indicating that PBMCs contributed to signaling and decreased proliferation of abnormal cells, avoiding the oncogenic role that cyclin D has when overexpressed.

Normally, p27 levels are increased in quiescent cells and fall rapidly after stimulation by mitogens. The downregulation of p27 has been linked to a worse prognosis for patients with several carcinomas, including oral squamous cell carcinoma, being directly or indirectly related to invasion and abnormal cell proliferation [31]. However, it is still not clear at which stage of carcinogenesis the downregulation of this protein occurs. Our findings indicate that the presence of PBMCs stimulates the activation of inhibitory mechanisms on cell proliferation and differentiation, although no differences in Ki-67 labeling rate were observed. Furthermore, this change was accompanied by changes in the levels of H3K9ac and γH2AX. DNA damage-induced cell cycle arrest occurs through several mechanisms. In this sense, DNA damage can trigger innate immune response, which leads to the production of type I interferons (IFN). The IFNs, in turn, activate a transcriptional response that leads to the upregulation of inflammatory cytokines and chemokines as well as other molecules with anti-tumor activities such as apoptosis-inducing proteins and natural killer (NK) cells [32,33]. Additionally, DNA damage can increase the expression of MHC molecules, which in turn can lead to recognition by the adaptive immune system and the generation of an antitumor response [34]. Furthermore, the presence of DNA damage can lead to the generation of neoantigens, which can be recognized by the immune system and lead to the production of tumor-specific antibodies and T-cell responses [35]. Thus, DNA damage can not only trigger an innate immune response but can also lead to an adaptive immune response.

The epithelial–mesenchymal transition (EMT) is a shift from epithelial to mesenchymal cell phenotype, which leads to loss or reduced expression of epithelial cell-specific markers and increased expression of mesenchymal cell-specific markers [36]. Despite being an embryonic development program, EMT is evoked during tumor progression, invasion, and metastasis. Tumor cells that undergo EMT acquire spindle-shaped morphology (fibroblast-like), showing significant loss of cell–cell contact and an increased expression of mesenchymal markers such as vimentin [37,38]. The results of our study suggest that PBMCs may participate in EMT events in dysplastic and neoplastic cells, inducing phenotypic changes that may favor progression to malignant transformation and invasion.

Inflammation plays a critical role in regeneration and repair. When a cell is exposed to pathogens or damage, pattern recognition receptors become activated, leading to an inflammatory response and the activation of transcriptional effectors. These trigger global changes in epigenetic enzymes, increasing DNA accessibility and expanding the cell’s transcriptional repertoire, allowing for phenotypic flexibility [39]. This is a crucial cellular adaptation to injury or invasion. We evaluated the contact of oral keratinocytes with PBMCs and observed decreased histone acetylation. In this condition, the DNA becomes more condensed, and therefore, less gene activation or even the silencing of some genes occurs. In the context of trauma/injury, the inflammatory cells migrate to the affected site and increase DNA accessibility, and thus, genes that were silenced start to be stimulated to promote the production of proteins necessary for cell repair. In our study, we probably induced a trauma when generating the wound using the p200 and when we added the inflammatory cells; they probably favored acetylation in order to decondense the chromatin and promote gene activation, favoring repair. This is different from what we expected, as a decrease in H3K9ac levels was identified by Western blot. However, in the scratch assay, we have the trauma factor, which was probably caused epigenetic plasticity, increasing gene susceptibility to assist repair.

The main limitation of this study is the use of a conventional two-dimensional monolayer co-culture model to evaluate the complex interactions between epithelial cells and their microenvironment. Thus, studies using more complex in vitro or animal models are needed to confirm the role of the immune cells and their influence on the pathogenesis and malignant transformation rates, since these may be possible therapeutic targets.

## 5. Conclusions

In conclusion, the presence of chronic inflammation associated to dysplastic lesions is capable of promoting epigenetic alterations, which in turn can favor the process of malignant transformation. It is important to identify the specific molecular pathways and mechanisms that are involved in the malignant transformation of oral lesions with epithelial dysplasia. This includes the epigenetic regulators which are involved in controlling gene expression and chromatin structure. Additionally, it is important to understand the downstream effects of targeting epigenetic regulators, such as the induction of apoptosis, proliferation, EMT, and immune modulation. Finally, it is necessary to understand the role of post-translational modifications, such as acetylation, in regulating the activity of epigenetic regulators and their downstream effects. By understanding the molecular mechanisms that dictate the outcome of targeting epigenetic regulators for epithelial dysplasia and inflammation, it will be possible to develop effective therapeutic strategies for the treatment of oral epithelial dysplasia.

## Figures and Tables

**Figure 1 jpm-13-00662-f001:**
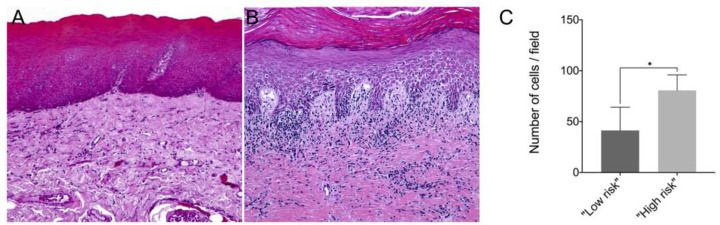
Representative image of hematoxylin–eosin (HE) staining from a (**A**) “low-risk” lesion biopsy and a (**B**) “high-risk” lesion biopsy; (**C**) Number of inflammatory cells per field in samples from “low-risk” and “high-risk” lesions. The mean number of immune cells counted per field in paraffin-embedded tissue samples of “low risk” and “high risk” lesions. * *p* < 0.05.

**Figure 2 jpm-13-00662-f002:**
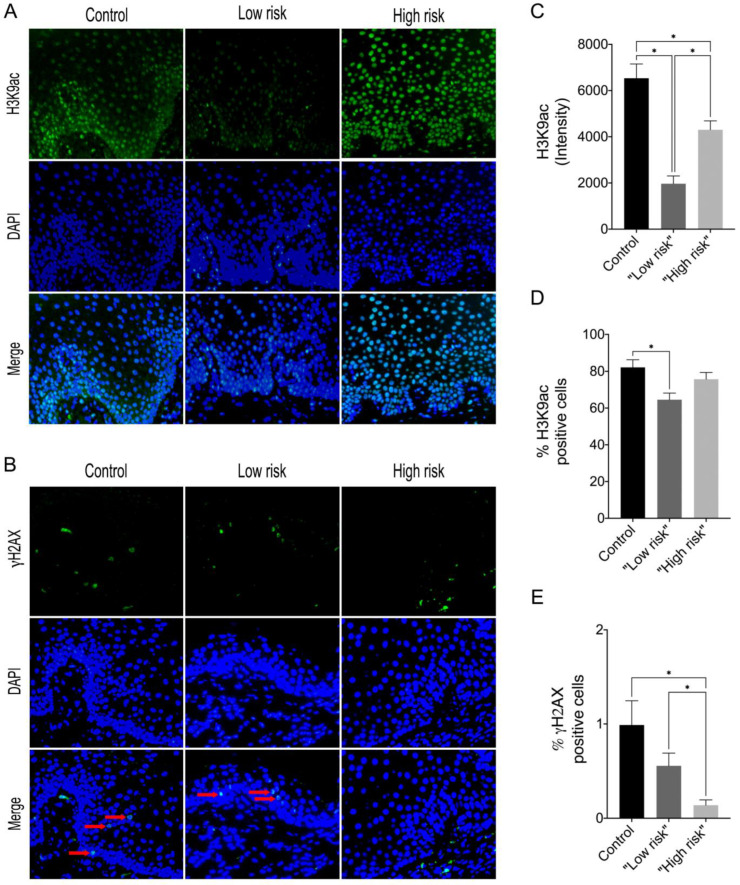
Representative images of immunofluorescence analysis for (**A**) H3K9ac and (**B**) γH2AX in control, “low-risk” and “high-risk” groups (Alexa Fluor 488, 40×) The red arrows indicate positive staining in the intraepithelial region. (**C**) Graphics of quantitative analysis of H3K9ac immunofluorescence intensity, (**D**) Percentage of the total number of H3K9ac positive cells (**E**) Percentage of the total number of γH2AX positive cells. * *p* < 0.05.

**Figure 3 jpm-13-00662-f003:**
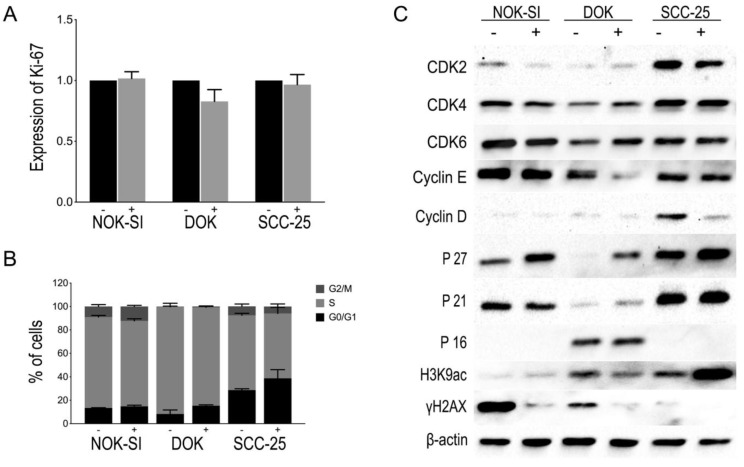
(**A**) Effect of PBMCs on the proliferation of NOK-SI, DOK, and SCC-25 cell lines. The graph represents the relative MIF of Ki-67 expression obtained from different experiments for each cell line in the absence (−) or presence (+) of PBMCs. No statistically significant differences were observed. (**B**) Influence of PBMCs on cell distribution of NOK-SI, DOK, and SCC-25 cell lines in the absence (−) or presence (+) of PBMC. No statistically significant differences were observed in the distribution of the cells in the different phases of the cell cycle. (**C**) Representative images of Western blot reactions with protein extract from NOK-SI, DOK, and SCC-25 cell lines in the absence (−) or presence (+) of PBMCs, showing the modulation of the expression of some proteins involved in the G1-S transition of the cell cycle and also slight changes in H3K9ac levels, except in SCC-25, and a decrease in γH2AX levels.

**Figure 4 jpm-13-00662-f004:**
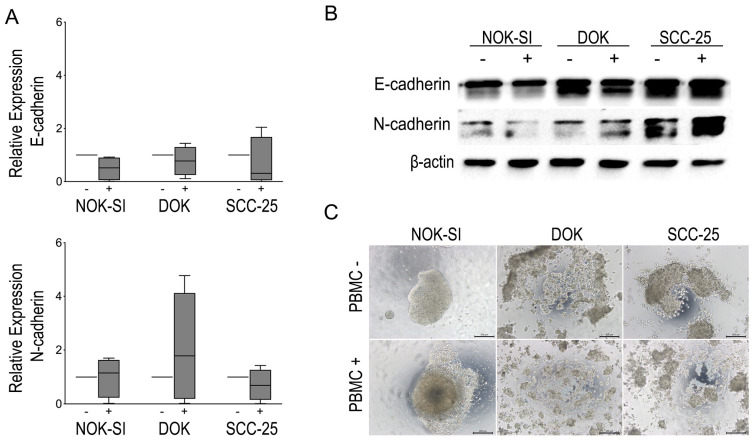
(**A**) Modulation of E-cadherin and N-cadherin expression evidenced by qRT-PCR. (**B**) Representative images of Western blot for EMT markers showing modulation of E-cadherin and N-cadherin in the different cell lines in the absence (−) and in the presence (+) of PBMCs for a period of 48 h. Each cell line in the absence (−) or presence (+) of PBMCs. (**C**) Hanging drop assay showing the pattern of spheroid formation in NOK-SI, DOK, and SCC-25 cell lines and the impact of PBMCs on the spheroid formation of each cell line.

**Figure 5 jpm-13-00662-f005:**
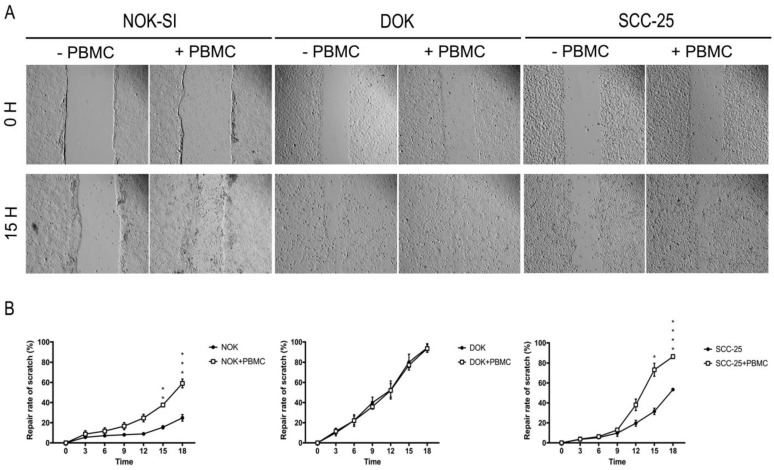
Influence of PBMCs on cell migration of NOK-SI, DOK, and SCC-25 cell lines. (**A**) Representative images of each condition at 0 and 15 h. (**B**) The presence of PBMCs was able to reduce the wound size in NOK and SCC-25 cell lines. * *p* < 0.05.

## Data Availability

Anonymized source data are available upon reasonable request.

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
