# Peer review of "Inflammatory Cells Can Alter the Levels of H3K9ac and γH2AX in Dysplastic Cells and Favor Tumor Phenotype"

_jpm, 2023, doi:10.3390/jpm13040662_

Round 1

Reviewer 1 Report

Comments are in the pdf

Author Response

RESPONSE TO REVIEWERS COMMENTS

Manuscript ID: jpm-2313735

Title: Inflammatory cells can alter the levels of H3K9ac and γH2AX in dysplastic cells and favor tumor phenotype

Authors: Camila de Oliveira Barbeiro, Darcy Fernandes, Mariana Paravani Palaçon, Rogerio Moraes Castilho, Luciana Yamamoto De Almeida, Andreia Bufalino *

Dear reviewer,

Thank you for your considerations.

  1. Keywords must be in alphabetical order

Response: We placed the keywords in alphabetical order (“co-culture; epithelial dysplasia; malignant transformation; peripheral blood mononuclear cell). The changes are highlighted in the text.

  1. Mention the stats

Response: We wrote in the text the statistics of the overall proportion of the malignant transformation rate among the OPMDs according to the article by Locca et al., 2020 (“Considering all OPMD, the overall rate of malignant transformation was 7.9% [2]). This reference was included in the text and in the reference list, and we also changed the numbering of the other references that were already in the text. The changes are highlighted in the text.

  1. This line required further context as it is hard to follow

Response: We rewrote the phrase in the text: “In view of these findings, the concept of cancer immuno-editing emerged. This comprises the elimination phase, in which the immune system recognizes and kills the potentially malignant cells; the equilibrium phase, in which an immune selection of tumor cells with reduced immunogenicity occurs; and the escape phase of tumor development, in which the immune system acts favoring tumor progression.” The changes are highlighted in the text.

  1. Avoid this term. Use potentially malignant cells

Response: We changed the term “pre-cancerous” to potentially malignant cells. The changes are highlighted in the text.

Reviewer 2 Report

Dear authors, this is a well designed study of inflammatory infiltration of dysplastic cells, and how this can affect histone acetylation (H3K9ac marker) and DNA damage. The only two things that I would like to request some clarification are

1. sample size is small, why did you choose this number? Any specific reason for that?

2. Increased repair rates in the PBMCs group are noted. Some more clarification in the discussion section could be given, as the epigenetic pathways can be complex.

3. Checklist for Reporting In-vitro Studies – CRIS Statement  could be implemented for standardizing the report's structure.

Author Response

RESPONSE TO REVIEWERS COMMENTS

Manuscript ID: jpm-2313735

Title: Inflammatory cells can alter the levels of H3K9ac and γH2AX in dysplastic cells and favor tumor phenotype

Authors: Camila de Oliveira Barbeiro, Darcy Fernandes, Mariana Paravani Palaçon, Rogerio Moraes Castilho, Luciana Yamamoto De Almeida, Andreia Bufalino *

Dear reviewer,

Thank you for your considerations.

  1. Sample size is small, why did you choose this number? Any specific reason for that?

Response: For this study, we calculated the sample size for all groups using the following parameters: minimum difference of means, standard deviation of the standard error, α=0.05, test power and number of treatments, which resulted in a minimum number of 4 samples for each group. We included this clarification in the text and the changes are highlighted in the text.

  1. Increased repair rates in the PBMCs group are noted. Some more clarification in the discussion section could be given, as the epigenetic pathways can be complex.

Response: Thank you for your note. We have included and clarified the results of the scratch assay in the discussion. We have added the following paragraph in the discussion section: “Inflammation plays a critical role in regeneration and repair. When a cell is exposed to pathogens or damage, pattern recognition receptors become activated, leading to an inflammatory response and activation of transcriptional effectors. These triggers global changes in epigenetic enzymes, increasing DNA accessibility and expanding the cell's transcriptional repertoire, allowing for phenotypic flexibility [39]. This is a crucial cellular adaptation to injury or invasion. We evaluated the contact of oral keratinocytes with PBMCs and observed decreased histone acetylation. In this condition, the DNA becomes more condensed and therefore less gene activation or even silencing of some genes occurs. In a context of trauma/injury, the inflammatory cells migrate to the affected site and in-crease DNA accessibility and thus genes that were silenced start to be stimulated to promote the production of proteins necessary for cell repair. In our study, we probably induced a trauma when generating the wound using the p200 and when we added the inflammatory cells, they probably favored acetylation in order to decondense the chromatin and promote gene activation, favoring repair. This is different from what we expected, as decrease in H3K9ac levels was identified by western blot. However, in the scratch assay we have the trauma factor, which probably caused an epigenetic plasticity, increasing gene susceptibility to assist repair.” The changes are highlighted in the text.

  1. Checklist for Reporting In-vitro Studies – CRIS Statement could be implemented for standardizing the report's structure.

Response: We have checked the items on the checklist for in-vitro study reports and added the information that was missing. The changes are highlighted in the text.